# Using camera-guided electrode microdrive navigation for precise 3D targeting of macaque brain sites

Max Arwed Crayen[1,2,3], Igor Kagan[1,4], Moein Esghaei[1,5], Dirk Hoehl[6], Uwe Thomas[6], Robert Prückl[7], Stefan Schaffelhofer[7], Stefan Treue[1,2,4,8] *

1 Cognitive Neuroscience Laboratory, German Primate Center, Goettingen, Lower Saxony, Germany, 2 Faculty of Biology and Psychology, Georg-August University, Goettingen, Lower Saxony, Germany, 3 International Max Planck Research School for Neurosciences, Georg-August University, Goettingen, Lower Saxony, Germany, 4 Leibniz ScienceCampus Primate Cognition, Goettingen, Lower Saxony, Germany, 5 School of Cognitive Sciences, Institute for Research in Fundamental Sciences (IPM), Tehran, Iran, 6 Thomas RECORDING GmbH, Giessen, Hesse, Germany, 7 cortEXplore GmbH, Linz, Upper Austria, Austria, 8 Bernstein Center for Computational Neuroscience, Goettingen, Lower Saxony, Germany

* treue@gwdg.de

**Data Availability Statement:** All relevant data are available from the Göttingen Research Online (GRO) Database (https://doi.org/10.25625/SUAMUR).

## Abstract

Spatial accuracy in electrophysiological investigations is paramount, as precise localization and reliable access to specific brain regions help the advancement of our understanding of the brain's complex neural activity. Here, we introduce a novel, multi camera-based, frameless neuronavigation technique for precise, 3-dimensional electrode positioning in awake monkeys. The investigation of neural functions in awake primates often requires stable access to the brain with thin and delicate recording electrodes. This is usually realized by implanting a chronic recording chamber onto the skull of the animal that allows direct access to the dura. Most recording and positioning techniques utilize this implanted recording chamber as a holder of the microdrive or to hold a grid. This in turn reduces the degrees of freedom in positioning. To solve this problem, we require innovative, flexible, but precise tools for neuronal recordings. We instead mount the electrode microdrive above the animal on an arch, equipped with a series of translational and rotational micromanipulators, allowing movements in all axes. Here, the positioning is controlled by infrared cameras tracking the location of the microdrive and the monkey, allowing precise and flexible trajectories. To verify the accuracy of this technique, we created iron deposits in the tissue that could be detected by MRI. Our results demonstrate a remarkable precision with the confirmed physical location of these deposits averaging less than 0.5 mm from their planned position. Pilot electrophysiological recordings additionally demonstrate the accuracy and flexibility of this method. Our innovative approach could significantly enhance the accuracy and flexibility of neural recordings, potentially catalyzing further advancements in neuroscientific research.

## Introduction

Recording extracellular potentials within the macaque cortex and subcortical structures provides vital insights into the complexity of brain functions, from cognitive processes to disease

**Funding:** This study was supported by a grant to ST (Deutsche Forschungsgemeinschaft (DFG, German Research Foundation) – Projektnummer 436260547). The funders had no role in study design, data collection and analysis, decision to publish, or preparation of the manuscript.

**Competing interests:** I have read the journal's policy and the authors of this manuscript have the following competing interests: DH & UT are technical director & CEO of Thomas RECORDING GmbH that developed the electrode microdrive system and the 'Precision Positioning System'. RP & SS are CFO & CEO of cortEXplore GmbH that developed the trajectory planning software and camera positioning system. MC, IK, ME & ST declare no conflict of interest.

mechanisms. A cornerstone of this research is the ability to precisely locate brain sites within cortical areas to obtain accurate recording and repeatable results. The reliable access to sites within one area allows the exploitation of many areas' functional organization, like the retinotopic organization in many visual areas for the study of complex cortical functions [1, 2]. This task, particularly for non-superficial sites, has been historically challenging due to the inherent complexities of the brain's folded structure and the limitations of available technologies. Current techniques, for example, often depend on the use of an implanted recording chamber to guide the recording trajectory either by holding an inset (grid), or by mounting a microdrive over it; restricting the electrode orientation to the chamber orientation [3–6]. This approach, while practical, does not grant enough flexibility to plan recordings in specific local regions based on the area's organization. As the exact functional anatomy of an area is often only revealed after the first recordings, it might be that the area of interest might not be easily accessible anymore with electrode orientations fixed to a grid's or chamber's orientation, or require elaborate angled grids [7, 8]. Additionally, these approaches for electrode navigation require detailed record keeping of the electrode microdrives' position set by micromanipulators and lead to yet larger accumulation of positioning errors without a possibility to verify the electrode trajectory independently. The emergence of multi-contact electrodes has further underscored the inflexibility of conventional techniques as now the angle of the approaching electrode(s) critically determines the number of accessible neuronal sites in the targeted area within one recording session, highlighting a pressing need for a more adaptable approach to electrode orientation. There have been earlier studies that used flexible, frameless approaches for neuronavigation in humans [9–12] and animals [13–17]. These approaches often used fiducial markers, rigidly attached, or implanted to the subjects, or intra-operative scans. Also, these techniques were evaluated in surgical environments under general anesthesia and were not tested for acute recordings in awake macaques. For the use in awake, behaving animals, a method without anesthesia is necessary.

Here, we introduce such an approach, using the novel neuronavigation system CORTEXPLORER SCI (cortEXplore GmbH, Linz, Austria) allowing camera-guided positioning of tools in animals and humans. We built a setup that combines this camera-guided navigation and positioning with an electrode microdrive. This allows us to overcome the constraints of limited flexibility in electrode trajectory planning for electrophysiological experiments, as this system functions independently from a recording chamber's orientation and provides precise and variable trajectory planning. Our study evaluates the accuracy of this system in denatured eggwhite (as a substitute for brain tissue), as well as in the cortex of two awake rhesus macaques.

In CORTEXPLORER SCI, a 3-dimensional representation of the target volume is created based on co-aligned MRI and CT scans. Thereby, electrode trajectories are defined in advance of a recording session by choosing the desired target- and entry points. The Mini Matrix electrode microdrive (Thomas RECORDING GmbH (TREC), Giessen, Germany; customized for cortEXplore compatibility) and the monkeys head are continuously tracked by cortEXplore's infrared camera system that allows precise evaluation of electrode positions relative to the target volume while advancing recording electrodes. We mount the electrode microdrive on a sturdy frame, rather than a recording chamber, allowing 3D navigation with increased degrees of freedom. To verify the accuracy of the microdrives and electrodes positions, we created local metal deposits at the recording site by applying an anodal current to quartz-glass-coated iron electrodes (Thomas RECORDING GmbH, Giessen, Germany) [18]. These deposit spots were detected in MRI scans and their relative offset to the planned target position were evaluated. Additionally, we piloted electrophysiological recordings that used this positioning technique to demonstrate the ease-of-use and effectiveness of this electrode positioning approach.

Our proposed technique aims to improve the planning and execution of brain recording and perturbation studies by not only addressing one of the biggest restrictions in current experimental protocols–the inflexibility of recording trajectories–but also improving the overall accuracy of targeted approaches to specific cortical and sub-cortical areas.

## Methods

### 3D tracking and positioning of electrodes and the target volume

We used a TREC 5 Tetrode Mini Matrix (Thomas RECORDING GmbH, Giessen, Germany) to drive the microelectrodes. This microdrive is capable of independently moving multiple microelectrodes or tetrodes, using a patented electronically controlled rubber tube system [19]. The use of transdural guide tubes to drive the electrodes made the removal of the *dura mater* unnecessary, which kept the natural immune barrier intact. The thin fiber electrodes have also been shown to minimize tissue strain compared to other electrodes [20]. Instead of mounting it to a skull-based recording chamber, limiting the degrees of freedom of movement, we used an arch (Thomas RECORDING GmbH, Giessen, Germany; Fig 1). We mounted the Mini Matrix' housing to the arch, using a combination of two rotational and three linear micromanipulators (3xLT 60–100, 1x GO 150-20-243, and 1x DT 65-M6; OWIS GmbH, Staufen i. Br., Germany), which allows fine angular adjustments along the rotation axis defined by the arch, fine tuning of the microdrive position in all three cardinal axes, and rotation around the electrode axis. In addition, the arch allows a coarse rotational adjustment in one axis. The arch's inner diameter is 79 cm and the target volume, either the brain tissue accessible by the chamber or the container holding the denatured egg-white, was positioned at the center point of the arch. The arch can be rotated along the axis defined by the two ends of the arch and the center point, and the height of the arch can be changed by sliding it along the two linear rails. The rails are mounted on an aluminum base plate which is bolted onto a 142 x 80 x 16 $cm^3$, 650 kg granite plate resting on 6 shock absorbers (cplusw GmbH, Hamburg, Germany) to minimize vibrational noise in the system. After positioning and penetration of the dura by the guide tubes, the electrodes are advanced into the tissue by the Mini Matrix. This setup allows flexible positioning of the electrodes in 3D with multiple degrees of freedom (Fig 1A). The system is closely modelled after the 'Precision Positioning System' by Thomas RECORDING, but we changed the arrangement of the linear and rotational micromanipulators to ensure that guide tubes could not accidentally penetrate tissue along an axis different from their longitudinal axis. Additionally, we use micromanipulators with an increased workspace as we need access to both hemispheres of a monkey's brain using only the linear and rotational manipulators mounted on the arch.

We equipped the TREC Mini Matrix with infrared-reflective (IR) markers to allow tracking of the guide tube axis and tip position with the CORTEXPLORER SCI system (cortEXplore GmbH, Linz, Austria). CORTEXPLORER SCI is a software and hardware system for frameless planning and neuronavigation (Fig 1B). Multiple IR-cameras (here four) are used to track the position of the Mini Matrix, a cranial reference geometry and a pointer tool. The cranial reference geometry is rigidly attached to the target (e.g., monkey's head) and serves for movement compensations during a procedure. The pointer tool was used for a two-step co-registration of the target to a corresponding digital representation derived from MRI and CT images. During the first point-set registration, we selected three arbitrary, well-spaced points on the outside surface of the monkey's chamber implant and sequentially touched them by the pointer tool. During the surface registration, the pointer tool was then used to sample over larger parts of the implant surface to refine the co-registration. Using all acquired data points from this tracking, the Root-Mean-Square error (RMS error) to the digital reference surface was calculated as

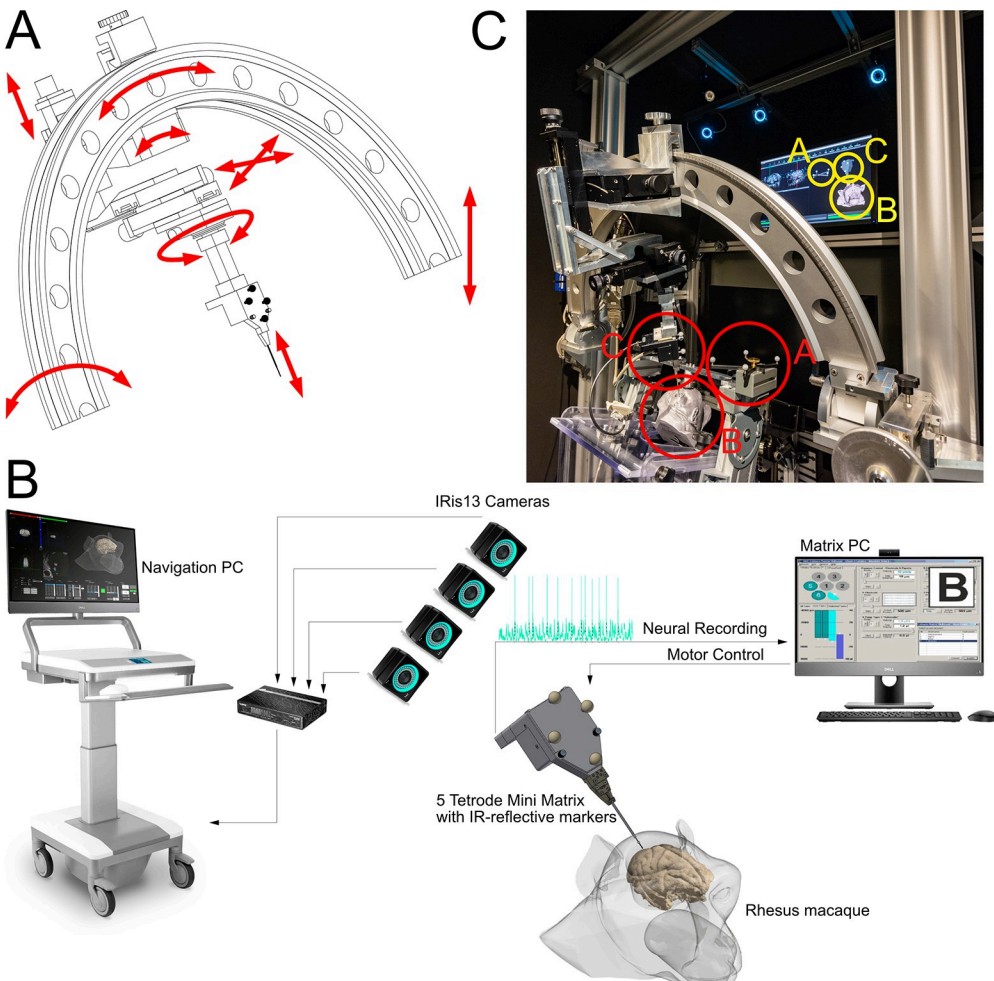

**Fig 1. Electrode microdrive positioning setup.** This setup allows movements in 3 translational and 3 rotational axes during online position tracking with IR-cameras. (A) The arch that is holding the microdrive is mounted on two vertical and parallel aluminum beams which allows to change its height. The arch can be rotated along the horizontal line connecting the vertical beams. A set of three linear micromanipulators is mounted on the arch and can be moved along the arch to rotate around the center point of the arch, where the target is placed. The linear micromanipulators can be used to adjust the 3 translational axes and to finally drive the tip of the guide tube down into the tissue. Lastly, the microdrive is connected to the linear micromanipulators on a table that is rotatable 360 degrees. (B) Layout of the CortEXplore System interacting with the TREC Microdrive. The CortEXplore Navigation PC is connected to IR-cameras that are tracking the monkey's head as well as the microdrive's position relative to each other. A recording trajectory is defined based on an experimental planning file that utilizes CT and MR imaging. The rotational and translational offsets of the realtime electrode trajectory are then displayed by the CortEXplore system, and the micromanipulators set accordingly to minimize the positioning error. Once the positioning error is minimized on all axes (excluding the penetration axis of the electrode), the TREC Matrix computer advances the electrode out of the guide tube along that axis to the desired target depth (adapted Figure, originally provided by CortEXplore GmbH). (C) Photograph of the setup with a monkey head plastic. Physical objects (red) and their representations in digital 3D space (yellow) are encircled. After co-registration, the stiff connection between the reference geometry A and the monkeys head B, allows positioning of the TREC Mini Matrix C relative to the monkey's anatomy.

a measure of accuracy. After computing the registration transform, surgical tools could not only be tracked, but also rendered relative to radiological data and the surgical plan in real-time.

Our experimental setup, combining the systems described in Fig 1A and 1B is described in Fig 1C.

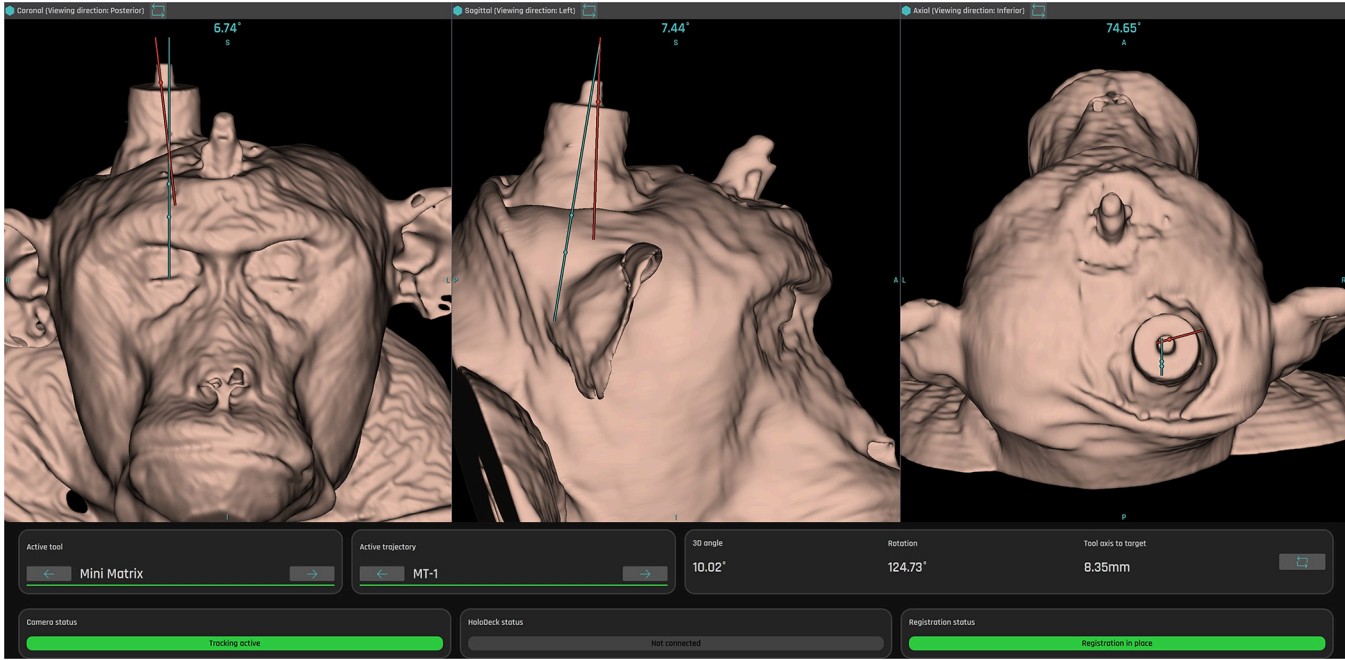

**Fig 2. Realtime electrode positioning is guided by CORTEXPLORER SCI.** The rotational offset of each axis is displayed on top of each axis section with the planned trajectory (blue) and realtime trajectory (red). The red dot along the realtime trajectory equals the guide tube tip of the MiniMatrix. The upper blue dot along the planned trajectory is the planned guide tube tip position, and the lower dot shows the electrode target position. The figure shows the realtime trajectory and positioning errors before the fine adjustment. Rotational offsets, displayed at the top of each window, are minimized first by adjusting the respective micromanipulators (rotational errors in this figure: coronal: 6.74˚, sagittal: 7.44˚, axial: 74.65˚). Afterwards, the translational offset of the realtime trajectory to the target location is minimized using the translational micromanipulators. Once the distance depicted in 'Tool axis to target' (here 8.35 mm) in the lower right corner is minimized (close to 0 mm), the microdrive is advanced into the tissue along the tool axis which is matching the guide tube.

Co-aligned magnetic resonance (MRI) and computer tomography (CT) imaging allowed for a 3-dimensional (3D) reconstruction of the skin, skull, and brain tissue of the monkey. Synthetic MRI- and CT-scans were generated based on the 3D model of the container holding denatured egg-white for planning and co-registration of the container during the experiment. After co-registration, the Mini Matrix was positioned based on pre-planned trajectories that allowed access to the planned target points. During this process, the CORTEXPLORER provided realtime feedback on the position of the Mini Matrix relative to the target volume as well as rotational and translational offsets to the planned trajectory (Fig 2). First, the Mini Matrix was positioned several centimeters above the target volume to allow positioning without restrictions by the target and then adjusted to minimize the rotational offset. Once the Mini Matrix' trajectory and the planned trajectory are parallel, the translation was performed to minimize the distance between the two axes. As a last step, the Mini Matrix was advanced along its axis into the target volume towards its planned position.

Once the Mini Matrix position was set, the distance from the guide tube tip to the target area that was calculated by CORTEXPLORER SCI was used to set the electrode penetration depth with the Mini Matrix Motor Control Unit.

## MRI scans

Anatomical MRI scans were performed for the monkey before the first iron deposition experiment to serve as a baseline reference. After the iron deposition, MRI scans were performed to locate the iron deposits in the target volume. The scans were performed in a 3T MRI (3T Magnetom Prisma, Siemens Medical Solutions, Erlangen). For the experiment in denatured egg-

white, the 3D model of the container used in the experiment was digitally converted into a synthetic MRI scan to serve as a reference for planning an evaluation of the results.

The scans from both monkeys were performed using a single receive RF coil with an 11 cm diameter (Loop11, Siemens Medical Solutions, Erlangen). Three T1-weighted images were taken at a spatial resolution of 0.5 x 0.5 x 0.5 mm$^3$ with 2.96/2700/850 ms of TE/TR/TI and 1 echo train length and averaged. Additionally, a T2-weighted scan, averaged across 4 repetitions, was performed at a spatial resolution of 0.25 x 0.25 x 1.00 mm$^3$ with 21/6000 ms of TE/TR and 7 echo train length.

The scans with the hard-boiled egg-white container were done using a coil with a 7 cm diameter (Loop7, Siemens Medical Solutions, Erlangen) using the same settings as during the monkey scans.

## CT scans

Before the experiment, a CT scan was done with both monkeys at a resolution of 0.35 x 0.35 x 1.00 mm$^3$. The scan was co-aligned with the corresponding MRI scans and later used to co-register the monkeys' heads with the CortEXplore system.

## Trajectory planning

Before each experiment, the electrode trajectories were defined in CORTEXPLORER SCI. A trajectory was defined by two points; entry and target. Based on T1 MRI images, areas of interest were defined as target and a corresponding entry was chosen just below the *dura mater* in a way that the trajectory was within the boundaries of the recording chamber. For the egg-white container, the synthetic MRI was used for planning target and entry points. In each experiment, the entry point is the planned final location of the guide tube tip, while the target point is the planned location of the electrode tip.

## Neural current application setup

The creation of iron deposits in the target volumes was performed using quartz-glass-coated iron electrodes (Thomas RECORDING GmbH, Giessen, Germany) that were driven into the target area by the TREC Mini Matrix. The electrodes had an outer shaft diameter of 80 μm. The Mini Matrix' 5 guide tubes were arranged in a cross layout and the outer diameter of each guide tube was 305 μm (Fig 3).

Once the electrodes were positioned in the target area, a WPI Stimulus Isolator (A 365R, World Precision Instruments, Sarasota, Florida, USA) was used to feed an anodal direct current of 4 μA into the electrode for 300 seconds to electrolytically deposit a very small amount of iron (~ 350 ng) in the surrounding brain tissue that can be detected by MRI. The application time and current settings were adapted from Koyano et al. 2011 [21]. The guide tube was acting as a ground for this procedure.

## Experiments

**Iron deposition in denatured egg-white.** A custom-designed 3D printed container was filled with egg-white and microwaved at 150W until the egg-white was denatured. The container was placed in the setup where the recording chamber of a monkey would be. In this experiment, a total of 5 quartz-glass-coated iron electrodes were each used for 2 iron depositions. The stimulation was done in two batches. After the initial co-registration of the container with CORTEXPLORER SCI, 4 stimulations with 2 electrodes were performed. Then the co-registration was reset and performed again, after which 6 stimulations with 3 electrodes were done. After the iron deposition, the container was immediately brought to the MRI

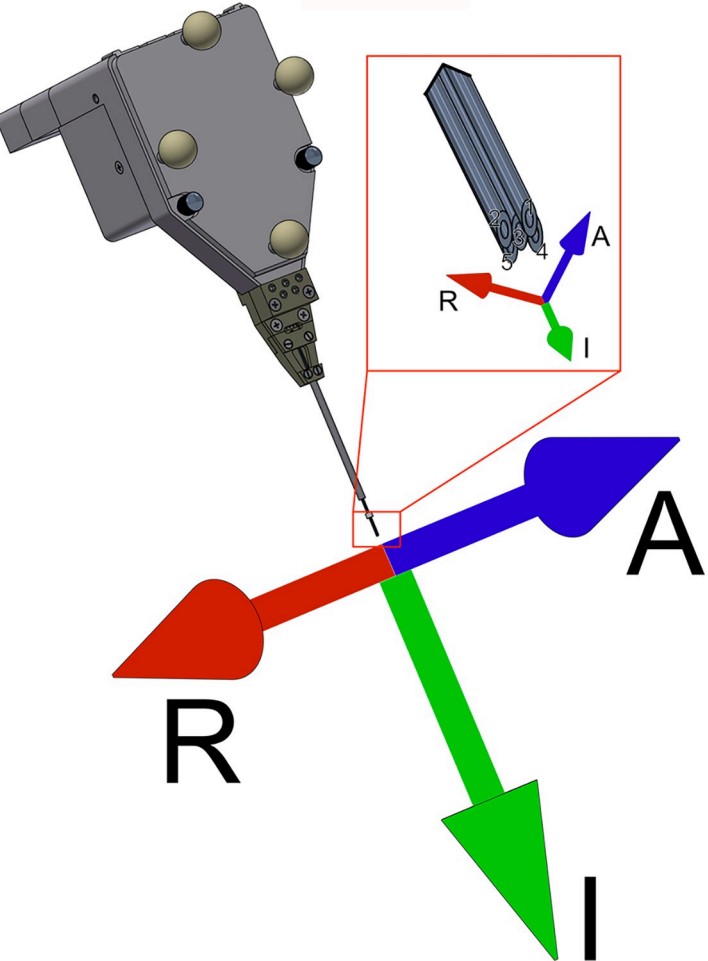

**Fig 3. TREC MiniMatrix guide tube layout and coordinate system for distance measurements.** The coordinate system used to report data in this study is based on the penetration axis of the TREC Mini Matrix microdrive's guide tubes. The coordinate system is centered around the tip of the central guide tube. The penetration axis of the guide tubes is set as the SI-axis of the coordinate system. The anterior direction A of the AP-axis is defined as the direction facing the cameras, identifiable by the 4 spherical IR-reflective markers. The right-side R of the RL-axis is results from the definition of the other two axes. The guide tubes were ordered in a cross shape, centered around guide tube 3.

scanner to perform T1- and T2-weighted scans for the localization of the iron deposits and the penetration marks left behind by the guide tube or electrode. We faced difficulties in denaturing the egg-white homogeneously while minimizing the number of air bubbles in the volume. The egg-white had to be denatured in the 3D-printed container and not in-prior to ensure that the egg-white does not move in the container while transporting it between the experimental setup and the MRI or also while driving the electrodes. Therefore, we could not boil the egg-white in advance and then put it into the container, which in turn would have led to minimized air bubbles. Due to microwaving, the egg-white did not denature homogeneously and was still more liquid in some parts of the volume than others. Denatured egg-white was chosen as a substrate after first experiments using agar at different concentrations were unsuccessful. The agar was created with Ringer-solution to mimic conditions similar to live tissue, was more homogenous in consistency and could be produced without any air bubbles. Unfortunately, it was not possible to reliably create and detect iron deposits in agar.

**Iron deposition in the brain of an awake rhesus monkey.** The baseline CT and MRI scans were performed 11 days before the iron deposition under general anesthesia using an MRI-compatible stereotaxic frame (Model 1430M, David Kopf Instruments, Tujunga, CA, USA). On the day of the iron deposition, the 5 electrode Mini Matrix was loaded with 3 iron electrodes and mounted on the arch. The monkey was brought in the setup and was co-registered to the CORTEXPLORER SCI system while performing his usual training for a different project. All electrodes were driven with the same co-registration of the CORTEXPLORER SCI system. The first electrode broke while being advanced out of the guide tube, presumably because the dura was not penetrated yet. The guide tube was then advanced further into the tissue to ensure full penetration of the dura. The second electrode was used for one iron deposition but broke while being advanced to the second target location. The third electrode was used for deposits at two different target locations. 7 days after the iron deposition, T1- and T2-weighted MRI scans were performed to locate the iron deposits and possible penetration marks in the dura and brain tissue.

**Mapping of MT receptive fields in an awake rhesus monkey after cortEXplore guided electrode positioning.** The baseline CT and MRI scans were performed under general anesthesia 3 months and 3 days, respectively, before the start of the recordings, using an MRI-compatible stereotaxic frame (Model 1430M, David Kopf Instruments, Tujunga, CA, USA). The recordings were performed in the middle temporal area (MT) in the context of an attention experiment. We mapped receptive field to further evaluate the accuracy of the electrode positioning method. The location of area MT in the monkey's individual anatomy was estimated by morphing the D99 monkey atlas brain cortex segmentation onto the monkey's MRI scan [22]. This was done using AFNIs animal_warper function [23, 24]. To identify the region of MT that is of interest we utilized MTs retinotopic organization [2, 25]. Based on the MRI, electrode tip positions were set to be roughly in layer 4 of the MT region that was of interest for the recording session. The electrode positioning was done identical to the iron deposition experiment. The electrodes were advanced to the depth planned in CORTEXPLORER SCI. Next, the recording of the neuronal activity was optimized via varying the penetration depth of the electrode. Once isolated, the receptive field mapping was performed. Based on the receptive field location and size, the next sessions' recording location was adjusted as to find neurons with ideal receptive field properties for the attention experiment.

## Data processing

We co-aligned the post-experimental MRI-scans with our experimental plan in CORTEXPLORER SCI, which allowed us to overlap the pre- and post-experimental scans as well as the planned electrode trajectories. To evaluate the accuracy of our electrode positioning, we used two metrics: First, we identified the iron deposits in the post-experimental scans by searching for their distortions in the MRI which appear as black voxels in T1 and T2 scans. Then we extracted the coordinates of these voxels with CORTEXPLORER and compared them to the coordinates of the electrode tips planned locations. Secondly, we located tissue effects along the trajectory of the guide tube and electrode, which provided us with a second data point for each trajectory and compared their distance to the planned electrode trajectory. We converted the coordinate system from a monkey-centric stereotaxic coordinate system that is used by CORTEXPLORER SCI to a Mini Matrix-centered coordinate system (Fig 3). Now the center of the coordinate system is the tip of the central guide tube. The Superoinferior (SI)-axis is identical to the electrode axis, while the AP-axis describes the offset of the data to the front or back. Lastly, the Right-Left (RL)-axis describes the offset to the left or right of the Mini Matrix. As the coordinate system was centered on the tip of the central guide tube, it was necessary to

correct for systematic offsets created when using surrounding guide tube. In these cases, the planned location of each iron deposit was adjusted for the offset created by the guide tube that was used to drive the electrode creating the iron deposit. The planned location of the guide tube penetration into the tissue was not adjusted for this offset as all 5 guide tubes were always advanced into the tissue simultaneously and thus induced no systematic offsets. The data were analyzed in MATLAB 2021a (Mathworks Inc., Natnick, Massachusetts, USA) and plotted using Pierre Morels gram plotting library for MATLAB [26].

## Behavioral training setup

The behavioral training of the monkeys was controlled by the open-source software MWorks (mworks-project.org) running on two Apple iMac computers (Apple Inc., Cupertino, CA, USA), a client and a server. Eye and gaze positions were recorded using an Eyelink 1000 system (SR-Research, Ottawa, ON, Canada). During recordings, the monkeys were seated in a custom-built primate chair and viewed a 31.5" IPS-LCD monitor (AG324UWS4R4B, AOC, Taipeh, Taiwan) from a distance of 49 cm in a dark room. The monitor had a resolution of 3840 x 2160 pixels, was set to a refresh rate of 120 Hz, and covered 70˚ x 40˚ of the visual field.

## Receptive field mapping

Receptive field mapping was performed by probing the right visual hemifield using a random dot pattern (RDP) stimulus, since the recording chamber was implanted over the left hemisphere. The RDP had a radius of 1.5 degrees of visual angle (dva), a dot density of 5 dots/dva$^2$ with a dot diameter of 0.2 dva and dots moving randomly (0% coherence) at a speed of 3 dva/s. For the attention experiment a circular stimulus with a diameter of 40 dva, centered around the fixation point was used. Only the area in the right visual hemifield within this stimulus was probed during the receptive field mapping. This resulted in 92 possible probe locations. The probe locations were not overlapping with each other. Each probe was presented for 300 ms and each location was repeated 10 times.

## Neural recording setup

For each recording session, one quartz-glass insulated platinum/tungsten microelectrode (Thomas RECORDING GmbH, Giessen, Germany) was advanced into the posterior wall of the superior temporal sulcus, targeting area MT using the TREC Mini Matrix also used for iron deposition experiments. Signals from the microelectrodes were pre-amplified by a factor of 19 and wide-band filtered from 0,034 Hz to 50 kHz by an analog pre-amplifier built into the Mini Matrix. Then the signals were amplified and recorded with a sampling rate of 40 kHz and 16-bit precision using an Omniplex acquisition system (Plexon, Dallas, TX, USA).

## Neural data processing

Action potentials "spikes" were identified using OfflineSorter V4 (Plexon, Dallas, TX, USA). The raw data was filtered with a 6-pole Bessel high-pass at a cut-off frequency of 250 Hz and spike waveforms were detected based on a manually determined threshold. These waveforms were then manually split into clusters based on different features as implemented in the software, including the first three principal components of the waveforms, the maximum and minimum voltage amplitude across the entire waveform length ("peak" and "valley"), or the waveform energy. For each recording, features were chosen according to the best separation between clusters. Note that for these recordings, there was mostly only one unit recorded so

that this procedure predominantly served the purpose to separate the signal from background noise.

## Animal welfare statement

Research with nonhuman primates represents a small but indispensable component of neuroscience research [27, 28]. The scientists in this study are aware and are committed to the responsibility they have in ensuring the best possible science with the least possible harm to the animals [29].

All animal procedures of this study have been approved by the responsible regional government office (Niedersaechsisches Landesamt fuer Verbraucherschutz und Lebensmittelsicherheit, LAVES) under the permit number 33.19-42502-04-18/2823. The animals were group-housed with other macaque monkeys in facilities of the German Primate Center in Goettingen, Germany in accordance with all applicable German and European regulations. The facility provides the animals with an enriched environment (including a multitude of toys and wooden structures [30, 31]), natural as well as artificial light, exceeding the size requirements of the European regulations, and access to outdoor space. We have established a comprehensive set of measures to ensure that the severity of our experimental procedures falls into the category of mild to moderate, according to the severity categorization of Annex VIII of the European Union's directive 2010/63/EU on the protection of animals used for scientific purposes [32]. Surgeries were performed aseptically under sevoflurane gas anesthesia using standard techniques, including appropriate perisurgical analgesia to minimize potential suffering. Additionally, the animals received Rimadyl to treat postoperative pain for at least three and up to seven days after the surgery and received prophylactic antibiotic treatment with Melsolute and Hostamox on the first and third day after surgery. The German Primate Center has several staff veterinarians who regularly monitor and examine the animals and consult on procedures. During the study, the animals had unrestricted access to food and fluid, except on the days where data were collected, or the animal was trained on their behavioral paradigm. On these days, the animals were allowed unlimited access to fluid through their performance in the behavioral paradigm. Here the animals received fluid rewards for every correctly performed trial. Throughout the study, the animals' psychological and veterinary welfare was monitored by the veterinarians, the animal facility staff, and the lab's scientists, all specialized in working with nonhuman primates. The two animals were healthy at the conclusion of our study and were subsequently used in other studies.

## Animals

Two male rhesus macaque monkeys (*Macaca mulatta*) contributed to this experiment. *Monkey toa* participated in the iron deposition experiment. *Monkey pan* participated in the experiment evaluating receptive field locations. The animals previously participated in and were meanwhile being trained for other research projects. Both animals were implanted with a customized titanium pin, orthopedically attached to the top of their skulls which allowed the minimization of head movements during experiments and recording chambers, which were implanted, based on magnetic resonance imaging (MRI), over the parietal lobe. The cylindrical recording chambers had an outer diameter of 24 mm; the inner diameter was 20 mm.

*Monkey toa* was 12 years old and weighed 9.6–10.0 kg during the 3-week period of data collection. During data collection, he was being trained for an attention task and was therefore already implanted with a titanium head holder on the top of his skull and a recording chamber over the right hemisphere (stereotactic coordinates: mediolateral (ML): 14 mm right; antero-posterior (AP): 2 mm posterior, angled 34 degrees to the anterior).

*Monkey pan* was 10 years old and weighed 7.1–7.6 kg during the 7 weeks period of data collection. The monkey was being recorded from for an attention task and was implanted with a titanium pin on the top of his skull and a recording chamber over the left hemisphere using the cortEXplore system (stereotactic coordinates: ML: 14.74 mm, A: 1.24 mm, angled 12.7 degrees to the lateral and 13.3 degrees to the posterior). The electrode positioning for the recording sessions was performed using the technique proposed here, and the receptive field mapping evaluated with respect to the monkey's individual anatomy to verify positioning accuracy of the method.

## Results

To establish the precision of our camera-based neuronavigation system, we conducted two experiments, one in denatured egg-white and the other in two rhesus monkeys. The electrodes were positioned using a system, designed to position a TREC Mini Matrix microdrive with 6 degrees of freedom, in combination with cortEXplore's neuronavigation system. We created small iron deposits by applying a direct electrical current to glass-coated iron electrodes in our target tissue, creating traces that are detectable using MRI. This allowed us to infer the electrode position in the tissue during the experiment, even after removing the electrodes. After co-aligning the post-experimental MRI images with our experimental plan in CORTEX-PLORER SCI, we extracted the coordinates of the iron deposits and compared them to the planned locations of the electrode tips that created them. Additionally, we determined the coordinates of tissue damages left behind from the guide tube and electrode penetrating the tissue. This allowed us to compare the planned penetration axis with the evaluated penetration axis. Here, we disregarded any depth information as it was not possible to reliably infer the guide tube tip position along the axis of penetration from the MRI. All offsets are reported relative to the Matrix coordinate system with the guide tube representing the z-axis and the center being the guide tube tips as shown in Fig 3.

Additionally, we performed receptive field mapping of neurons recorded in cortical area MT after positioning electrodes using CORTEXPLORER SCI. The cortex was segmented to establish the position of MT in the MRI scans co-aligned in CORTEXPLORER SCI. This enabled a precise planning of recording locations, targeting the experiment-specific functional region of MT already during the first recording sessions.

### Iron deposition in denatured egg-white

First, we performed experiments in denatured egg-white as it allowed us to test, optimize and estimate the precision of our technique before any animal experiments, reducing the need of extensive evaluations in such animals. The electrode trajectories were planned based on the CAD model of the 3D-printed container that held the denatured egg-white. MRI scans conducted after the experiment were then co-aligned to the model in CORTEXPLORER SCI. We measured the coordinates of our iron deposits (red dashed squares) as well as penetration marks of the guide tube and electrodes into the denatured egg-white (blue dashed squares) after the experiment (S1 Fig). The planned positions of the electrode and guide tube are indicated as red and blue dots respectively. The guide tube was driven into the denatured egg-white until it was well penetrated into the volume to ensure proper grounding of the electrode. Therefore, the guide tubes were sometimes penetrating deeper into the tissue than originally planned. The electrodes were advanced 5.8 mm—10.9 mm from the guide tube. In the example of S1A Fig, it is even possible to identify the trajectory of the electrode, connecting the guide tube tip and iron deposit. 10 iron deposits were created and could be located in the MRI in the experiments in denatured egg-white. Fig 5A illustrates the offsets of each iron deposit to its

planned location in each dimension, as well as the 2-dimensional distance to the planned electrode axis and the three-dimensional distance. Additionally, the figure depicts the distance of penetration marks visible in the denatured egg-white and their respective distances to the ideal electrode penetration axis.

For iron deposits, the offsets in the RL-, AP- and SI-axes were -0.10 mm, 0.04 mm, and 0.78 mm respectively (std = 0.35 mm, 0.35 mm, and 0.90 mm, respectively). This results in average distances of the iron deposits to their intended electrode axis of 0.44 mm with a standard deviation of 0.21 mm. As the distances in the SI-axis were notably higher than in the other two axes, this resulted in a three-dimensional mean deviation of 1.14 mm (std = 0.55 mm) to the target location.

Damage that was created by the guide tube or electrode penetration could also be identified. The distances of these penetration marks to their ideal axis were 0.03 mm and 0.14 mm in the RL- and AP-axis (std = 0.27 mm and 0.41 mm), and a 2-dimensional 0.44 mm (std = 0.23 mm) distance of the penetration marks to their ideal penetration axes.

## Iron deposition in a rhesus monkey

After the techniques first evaluation in denatured egg-white, we conducted the same procedure in a rhesus macaque to verify the accuracy in a typical use case. CT and MRI scans were conducted as baseline scans for the co-registration to the camera system and planning of electrode trajectories in the monkeys' brain. We created iron deposits at the planned positions in the brain tissue of a rhesus macaque using quartz glass-coated iron electrodes and detected the iron deposits using scans conducted after the experiments. To evaluate the accuracy of this positioning method, we compared the coordinates of the localized iron deposits to their planned position in MRI images as shown in Fig 4A–4D. Additionally, we evaluated the location of tissue damage caused by the guide tubes and penetrating electrode in the dura and brain areas that were more superficial than the target site to calculate their offset to the ideal axis of the electrode and guide tube trajectory. The guide tube was driven into the tissue until it just penetrated through the dura. The electrodes were advanced 6 mm—8.2 mm from their respective guide tubes.

In experiment 3, iron deposits were created and located in the MRI images. The distances from the evaluated location to their planned position as well as penetration marks that could be spotted along the penetration axis of the electrode are plotted in Fig 5B. For iron deposits, the offsets in the RL-, AP- and SI-axes were -0.08 mm, 0.10 mm, and -0.05 mm (std = 0.36 mm, 0.17 mm, and 0.34 mm), respectively. This results in average distances of the iron deposits to their intended electrode axis of 0.33 mm (std = 0.12 mm). As the distances in the SI-axis were higher as in the other two axes, this resulted in a mean deviation of 0.43 mm (std = 0.15 mm) in three-dimensional space to the target location.

Given that the MRI-scanning was performed within a few days of the electrode insertion the damage that was created by the guide tube or electrode penetration was still detectable. The distances of these penetration marks to their ideal axis were 0.06 mm and -0.16 mm in the RL- and AP-axis (std = 0.24 mm and 0.31 mm), and a 2-dimensional 0.35 mm (std = 0.11 mm) distance of the penetration marks to their ideal penetration axes.

The localization of iron deposits in denatured egg-white showed larger offsets in the SI-axis than for other axes. To investigate whether this offset was consistent for different guide tubes, we grouped the offsets of the iron deposits to their planned position by the guide tube that was holding the electrode. There we show that the offset is larger, meaning the iron deposits were located more superior than intended, when the guide tube used was one of the surrounding guides 1, 2, 4 or 5 (S2 Fig). Deposits created using the central guide 3 did not show such a large offset.

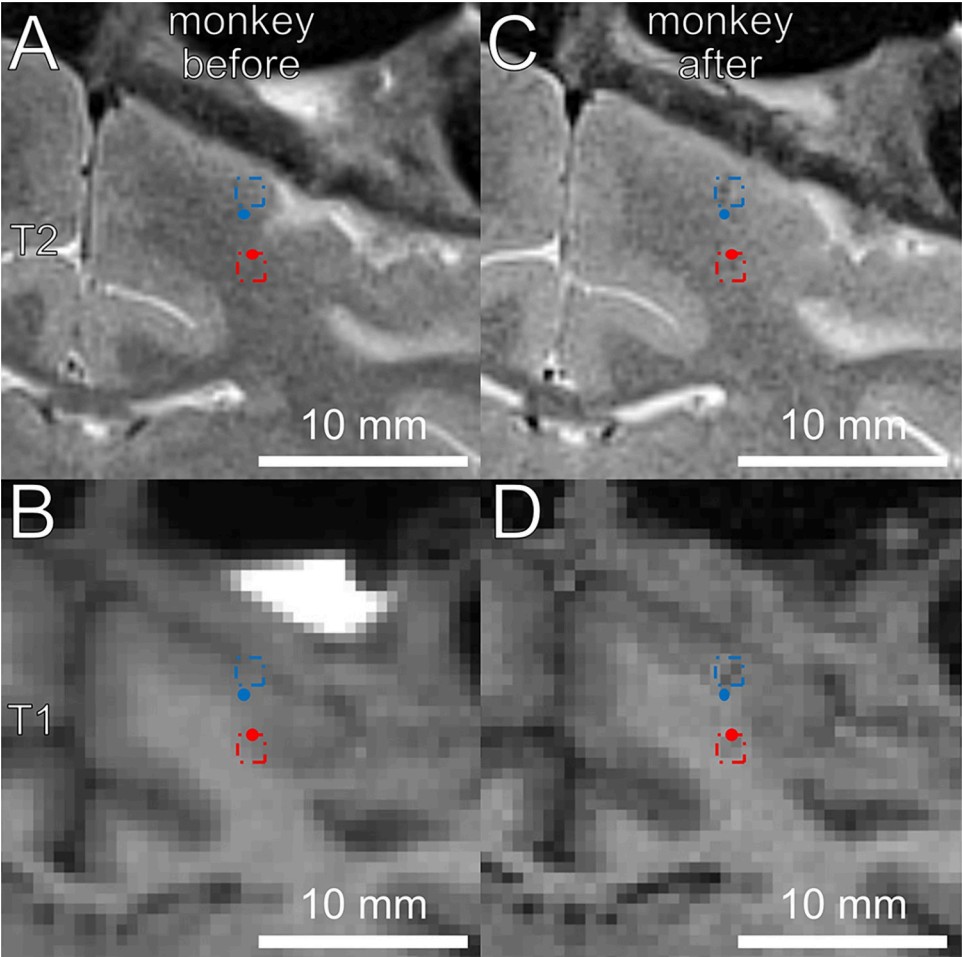

**Fig 4. T1- and T2-weighted MRI images of the monkey.** Planned (dots) and evaluated (squares) locations of the guide tube penetration entry point and iron deposits are added in blue and red, respectively. Due to the higher resolution and signal quality, T2-imaging was used for precise localization of iron deposits and guide tube penetration marks. A & C are T2-weighted images; B & D are T1-weighted images. A & B are images taken before the experiment; C & D were taken after the iron deposition.

The same analysis was repeated for the data collected in the monkey (S3 Fig), where such a large offset as in the first experiment were not observed in the SI-axis.

Besides the cardinal accuracy, the angular accuracy is of importance as well. We observed average angular errors of 3.02 degrees and 6.14 degrees in denatured egg-white and monkeys respectively with standard deviations of 1.87 degrees and 2.52 degrees (S4 Fig). Additionally, we calculated the correlation between the three-dimensional distance between the planned and evaluated iron deposit locations and the electrode penetration depth (S5 Fig). Our analysis shows no significant correlation (Pearson RHO = -0.17, p-value = 0.63; Spearman RHO = -0.28, p-value = 0.43).

### Receptive field mapping in macaque cortex area MT

Targeting specific regions within a brain area can be difficult and take multiple attempts until the ideal parameters for the positioning of the microdrive are found. To demonstrate the much simpler targeting of brains structures with this technique, we performed multiple

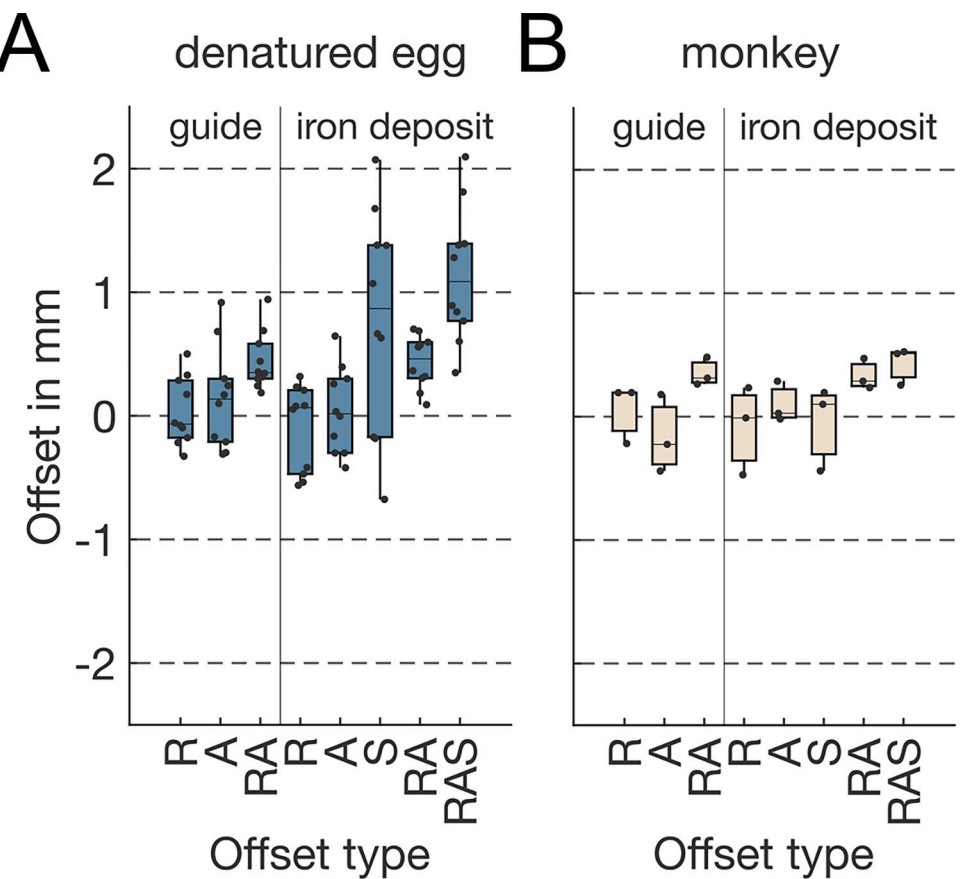

**Fig 5.** Translational error of the planned to physical location of iron deposits and penetration marks in the denatured egg-white container (A) and monkey (B) in mm. R is the offset to the right, A to the anterior and RA the 2-dimensional distance in the RA-plane. S is the offset in the superior direction and RAS depicts the 3-dimensional distance.

receptive field mappings in visual cortex area MT that is known to process linear motion direction information [33–35]. MT was rendered in CORTEXPLORER SCI after performing a cortex segmentation based on the D99 monkey atlas, allowing the planning of recording trajectories in the area. Considering the retinotopic organization of MT, we initially chose trajectories in the central region of MT which should allow access to neurons with receptive fields that have an eccentricity beyond 6 dva [2, 25]. The electrodes were first advanced to the planned depth, then single units were isolated by advancing and retracting the electrodes until a responsive unit was found. In Fig 6A–6C we show the recording trajectories of 4 example electrodes of different sessions in horizontal MRI slices and a 3D rendering. The yellow shaded area is the estimated location of MT based on the D99 monkey atlas morphed onto the animal's scan. When recording from central MT locations (Position 5) we observed relatively small receptive fields within 5 degrees of eccentricity, while the size grew larger and the position more eccentric, the more medial the recording locations were (Position 11, 12, 19) (Fig 6D). Furthermore, the receptive fields were wandering from the upper hemifield to the lower hemifield with deeper, more inferior recording locations. This is in line with the known functional anatomy of MT [2, 25, 35]. The recording location was chosen by first setting the electrode's penetration depth to the target location provided by CORTEXPLORER SCI. Then the depth was adjusted until a direction-selective neuron was found. In 18 recording sessions the

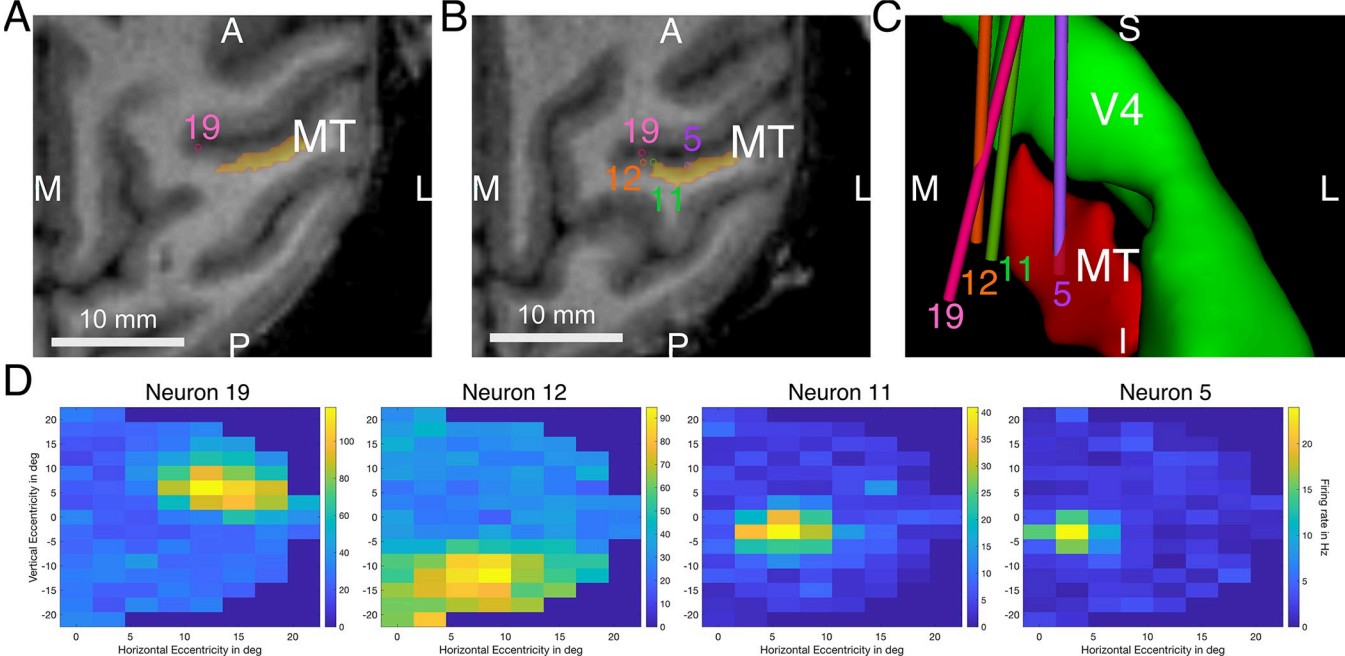

**Fig 6. Example recording trajectories and the respective receptive field maps of the recorded neurons.** (A). The recording location of neuron 19 in the bend of the wall of the superior temporal sulcus. This slice is more inferior than slice B. A = anterior, P = posterior, M = medial, L = left. (B). The recording locations of neurons 5, 11 and 12 as well as the location of electrode 19 passing through this slice. Area MT is rendered in orange based on warping of the D99 monkey atlas onto this monkey's brain. (C). A 3D rendering of the 4 electrode trajectories (colored cylinders) and Area MT and V4. Viewing direction is from anterior to posterior. (D). Receptive field maps of neurons 5, 11, 12 and 19.

recording depth of the electrode was on average 297 μm ±746 μm deeper than the CORTEX-PLORER SCI suggested depth. The total penetration depth was between 3366 μm and 11135 μm per electrode. In two of the recording sessions, no direction-selective neuron could be detected.

## Discussion

In this study, we demonstrated a novel technique that allows camera-guided 3-dimensional positioning of electrodes in the brain of rhesus macaques in real-time, observing an electrode positioning within approximately 0.4 mm of their target location. We did not restrict the electrode orientation to the chamber orientation by using an inset (grid), or by mounting the microdrive on it. With the increasing availability of multi-contact electrodes, such an inflexibility of the electrode orientation becomes a significant constraint. Here, we mounted the microdrive on an elevated arch which allowed 3-dimensional movement and positioning of the electrode trajectory, while simultaneously utilizing novel camera-guided navigation system giving realtime feedback on the microdrive's position relative to the target trajectory in the brain. This new method allows more degrees of freedom for trajectory planning and flexible changes, for example in cases where blood vessels restrict the access to a target area. Importantly, this method is not subject to accumulation of errors due to multiple micromanipulators setting the position of the microdrive holding the electrode as the position of the microdrive is directly tracked by IR-cameras. Though the fixation of the monkey's head is necessary, it is not required to precisely replicate the monkey's head position across recording sessions as it is often done in other stereotactic systems. During the recording or surgical procedures using our technique, the only requirement is that the monkey's head is in a stiff connection to the

positioning system. The cameras then automatically establish a stereotactic coordinate system for neuronavigation during the co-registration with CORTEXPLORER SCI.

The positioning error we observed in our experiment is lower than or equal to previously reported positioning techniques which utilized frame-based or frame-less methods [9–17]. The positioning errors in these techniques were often ranging between 1.2 mm—2.7 mm in humans [9–12] and animals [13–16]. In a recent study in NHPs, a frameless technique utilized the cranial cap screws that were drilled into the skull of a monkey during surgery as fiducial markers for neuronavigation [17]. As these screws are easy to access after a surgery and are a distinct landmark, their function as fiducial markers is ideal for repeated co-registration of the NHPs during neuronavigation techniques. The downside is the necessity of at least two surgeries as the cranial cap with screws first needs to be implanted, before the second neuronavigation guided surgery can be performed after an intermediate CT scan. Our technique allows the implantation of a cranial cap or recording chambers and electrode arrays in the same surgery since the registration of the monkey is not based on artificial fiducial markers. Instead, CORTEXPLORER allows to scan anatomical surfaces, such as the skull or the skin to align them with their corresponding virtual representations. This offers the potential of a reduced number of surgeries. This technique is also not limited to animals without implants. CT scans with already existing implants can also be used during the co-registration, further improving the positioning as implants are distinctly shaped landmarks that hardly change their shape over time. Importantly, cortEXplore-enabled tracking is possible with any implant and no specific adjustments are required to compatibility with this system.

We conducted this experiment first in a proxy made from denatured egg-white, but later also in live animals. This was done to allow testing and prototyping of the technique prior to the first animal experiments and thus reduce the demand for animal experiments according to the 3R principles. Denatured egg-white shares some characteristics of live brain tissue, but also has some obvious limitations. We chose this substrate as it allowed a reliable production of iron deposits with our technique.

When creating iron deposits in egg-white, we observed a relatively large variance when comparing the planned and evaluated locations. While the error was only 0.33 mm, when comparing the offset of the iron deposit to its creating electrodes' penetration axis, we observed a large variance along the penetration axis. This offset might have been caused by a variety of factors. There were differences in how the 0 position of electrodes were adjusted during the two different experiments. In the experiment with egg-white, each electrode was advanced to the opening of its individual guide tube, where the 0 position was then set. Unfortunately, it was not considered that the guide tubes were of different lengths, with the central guide tube (number 3) being the longest. The surrounding guide tubes 1,2, 4 and 5 were 500–1000 µm shorter. This guide tube length layout was chosen to archive a sequential guide tube penetration through the dura, but this offset was not accounted for as the digital model of the microdrive in CORTEXPLORER SCI was assuming all guides had the same length as guide tube 3. Thus, the evaluated locations of iron deposits created by guide tubes 1, 2, 4 and 5 were higher than planned. This can be observed when grouping the offsets by guide tube (S2 Fig). In the monkey experiment, we avoided this problem by advancing all electrodes to the height of guide 3's opening and setting all their 0 positions at this point (S3 Fig). Then the electrodes were retracted into their guide tubes so that they were not damaged while advancing through the dura. Later versions of the software also allow to calibrate tip positions of electrodes and thus, update the digital model to closer resemble the physical electrode layout. Additionally, there might have been movement of the iron deposits within the substrate. Iron that was released into the substrate would move along the electrical field towards the grounding guide tube, if possible. The previously discussed varying denaturation grade of the egg-white might

have allowed such movement and thus, located the iron deposits more superior in less denatured parts of the egg-white volume. In our and other studies involving the creation of metal deposits in live monkey cortex such a phenomenon was not observed yet [21, 36].

Overall, even when considering this error, we still observe a higher or equal positioning accuracy compared to recent other approaches [9–17]. After the refinement of the guide tube tip positioning and electrode depth calibration for the monkey experiment, we observed even lower errors in the positioning of our electrodes. The electrode tip positions were all aligned to the opening of the centered guide tube that was the most advanced, which resulted in physical electrode tip positions being more accurately matching the software's electrode tip position estimate that is based on the opening of that guide.

When evaluating this method, it is not only of interest to assess the translational offset, of the iron deposits and guide tube penetration locations to their planned locations. The angular offset of the penetration axis is of high interest as well. However, with penetration depths only varying between 5.8 mm and 10.9 mm, the two locations defining the axis were very close together. A shift of only the target location by 0.5 mm would already result in an up to 5 degrees increase of the angular error. We relied on MRI with a 0.5 mm isometric voxel-size for localization, so the angular errors are not considered to be too reliable, as a single voxel shift of the signal would already create a large angular error. For our data, the average errors were ~3 degrees in egg-white and ~6 degrees in monkeys (S4 Fig). As a single-voxel shift of the signal would thus result in errors equal or larger than the observed angular error, the observed errors might be strongly biased by measurement errors caused by the MRI's resolution and the real angular errors could be less than the reported ones. The lack of correlation between the penetration depth and three-dimensional offset (S5 Fig), further supports the hypothesis that the angular error in our approach is negligible. Only an increasing offset of the planned to evaluated target locations with increasing penetration depths would have pointed to a lower angular accuracy of this method. Still, as the penetration depths were at most 10.9 mm future experiments will have to further evaluate the angular precision when targeting deeper brain areas.

There are multiple relevant sources of error that affect the accuracy of this method. First, we must consider the physical error of the cameras when tracking the marker-equipped tools. According to the manufacturer cortEXplore, tracking under ideal conditions can be performed with errors below 200 μm. There is also the error between the physical objects and their digitally represented model counterparts that are used for planning and conducting the experiments in CORTEXPLORER SCI. It is difficult to measure this error directly and it is therefore of high importance to manufacture and assemble all parts with a high precision.

Importantly, the system does not require precise readings of micromanipulators that control the position of the Mini Matrix microdrive, as the system directly tracks the position of the microdrive, which allows much more flexible positioning processes. Still, a small positioning error remains during the manual positioning of the tools, when even small adjustments of the micromanipulators do not improve the positioning anymore. But this error is measured by the camera and the experimental plans or analysis can be corrected accordingly.

The quality of the co-alignment of the CT and MRI images is of course of high importance. As the CT is used for the co-registration to the physical objects and the MRI is used for planning, any offset between the CT and MRI would result in a physical offset of the tool to the planned target location in experiments.

The resolution and quality of MRI and CT scans thus strongly influence the positioning accuracy, but they are also used in other techniques, and as such play a less important role in the evaluation of error sources of this system. But they can be of interest when discussing the overall accuracy of this method. With the positioning error being evaluated around 0.33 mm–0.5 mm, we must consider the resolution of the MRI and CT imaging that the trajectory

planning was based on. T1-weighted MRI had isotropic voxel sizes of 0.5 mm, while T2-weighted MRI was 0.25 x 0.25 x 1.0 mm$^3$, with CT imaging voxel-sizes of 0.35 x 0.35 x 1.0 mm$^3$. The margin observed in this experiment is in the same range of the voxel-sizes we used for the planning and evaluation of this experiment. The actual accuracy of this method might be higher, but it is not possible to quantify it with the technical possibilities available to us.

Using iron deposits for the detection of electrode positions allowed us to use MRI for both, the baseline scan of the anatomy for the planning of the experiment, and the evaluation scan to detect the iron deposits. Compared to other techniques, like post-mortem histology, this approach has two major advantages. Firstly, there is no need to sacrifice or wait for the end of life of an animal for this experiment. Secondly, the co-alignment of the two MRI scans can be done with an higher accuracy compared to poor matching of a mixture of histology, CT and MRI images which in itself would introduce a localization error [37].

The spatial resolution we have observed in our experiments allows targeting of cortical areas with an error lower than 0.5 mm to the target location. It is thought that cortical columns, for example in area MT, have about the same diameter in the cortex [38–40]. Reliable access to the same cortical columns over several sessions are therefore possible, which can also be particularly interesting for microinjection experiments after recording from a given functional area. This is especially relevant in non-superficial sites like MT, which have always been more difficult to target than superficial cortex areas. Improving the targeting of brain sites has been an important topic, especially since neuroimaging-based brain perturbation studies have been established as a method to study causal mechanisms of behavioral and neural processes [41].

We applied this method to access macaque cortical area MT for an attention project in a second monkey. To evaluate accuracy of the electrode positioning in this context is of course much more difficult as we only have indirect ways of measurement. There are two important factors we can evaluate. First, the planned penetration depth compared to the actual recording location of a neuron. The planned recording location was set to target approximately layer 4 in the middle of the cortical sheet. Assuming perpendicular recordings, the cortex thickness of MT would be less than 2 mm [42]. Of course, it needs to be considered that our recording trajectories were not perpendicular to the cortex, but angled, so the depth range for recordings increases. Still, we have found neurons on average within 300 μm to the planned location, which suggests that electrode positioning using this method is very accurate. Second, we could also interpret the receptive field locations and sizes of the recorded neurons and compare them to the ideal retinotopic organization of MT in the literature [2, 25]. Here we found that the recorded neurons' receptive field profiles look as suggested by previous studies, although we found MT neurons with more eccentric receptive fields to be located more medial than it is rendered in the D99 monkey atlas [22]. Simply the fact that motion direction responsive neurons could be found starting with the very first recording session and allowed 18 successful recordings within the first 20 recording sessions, clearly indicates that accessing target areas using cortEXplore's system is a user-friendly and reliable approach.

In conclusion, our study introduces a flexible, user-friendly neuronavigation technique that revolutionizes three-dimensional electrode tip positioning and track planning in the rhesus macaque's brain. This method paves the way for a new era in the study of brain function by providing enhanced accuracy and flexibility compared to traditional methods, permitting independent access to targets from chamber-based grids or chamber-mounted microdrives. It proves particularly advantageous for multi-contact and multi-electrode approaches, enabling efficient planning and precise positioning for orthogonal and parallel recordings in identical cortical layers. Its superior accuracy complements the use of microinjection pipettes and other similar techniques, which often lack immediate positional feedback during the experiment.

Implementable even in animals with pre-existing implants, our novel technique improves the precision of electrode position detection, promising major strides in brain function studies, precise targeting of specific cortical areas, and improved facilitation of microinjection experiments. Looking forward, additional research should probe this method's aptitude for deeper brain targeting and its potential utility in neuroimaging-based brain perturbation studies.

## Supporting information

**S1 Fig. T1- and T2-weighted MRI images of the denatured egg-white.** Planned (dots) and evaluated (squares) locations of the guide tube penetration entry point and iron deposits are added in blue and red, respectively. Due to the higher resolution and signal quality, T2-imaging was used for precise localization of iron deposits and guide tube penetration marks. In A, the electrode tract connecting the two squares can be identified. A is a T2-weighted image; B is a T1-weighted image. A & B were taken after the iron deposition.
(TIF)

**S2 Fig. Translational error of the planned to physical location of iron deposits grouped by guide tube in the denatured egg-white container.** R is the offset to the right, A to the anterior and RA the 2-dimensional distance in the RA-plane. S is the offset in the superior direction and RAS depicts the 3-dimensional distance.
(TIF)

**S3 Fig. Translational error of the planned to physical location of iron deposits grouped by guide tube in monkey.** R is the offset to the right, A to the anterior and RA the 2-dimensional distance in the RA-plane. S is the offset in the superior direction and RAS depicts the 3-dimensional distance.
(TIF)

**S4 Fig. Angular errors of the planned to physical trajectories in the denatured egg-white container and monkey.**
(TIF)

**S5 Fig. Correlation between penetration depth and angular error in the denatured egg-white container (Pearson RHO: -0.1741, P: 0.6305; Spearman RHO: -0.2848, P: 0.4274).**
(TIF)

## Acknowledgments

We thank Klaus Heisig, Torsten Töteberg and Johannes Oehne for their support in the construction and manufacturing of large parts of the recording setup. We thank Sina Plümer, Leonore Burchardt, Daniela Lazzarini, and Akshay Edathodatil for their support with the monkey handling and conduction of the experiments.

## Author Contributions

**Conceptualization:** Max Arwed Crayen, Moein Esghaei, Dirk Hoehl, Uwe Thomas, Robert Prückl, Stefan Schaffelhofer, Stefan Treue.

**Data curation:** Max Arwed Crayen, Igor Kagan.

**Formal analysis:** Max Arwed Crayen.

**Funding acquisition:** Stefan Treue.

**Investigation:** Max Arwed Crayen, Igor Kagan, Stefan Treue.

**Methodology:** Max Arwed Crayen, Igor Kagan, Moein Esghaei, Dirk Hoehl, Robert Prückl, Stefan Treue.

**Project administration:** Moein Esghaei, Stefan Treue.

**Resources:** Dirk Hoehl, Uwe Thomas, Robert Prückl, Stefan Schaffelhofer, Stefan Treue.

**Software:** Max Arwed Crayen, Dirk Hoehl, Uwe Thomas, Robert Prückl, Stefan Schaffelhofer.

**Supervision:** Igor Kagan, Moein Esghaei, Stefan Treue.

**Validation:** Max Arwed Crayen.

**Visualization:** Max Arwed Crayen.

**Writing – original draft:** Max Arwed Crayen.

**Writing – review & editing:** Max Arwed Crayen, Igor Kagan, Moein Esghaei, Dirk Hoehl, Uwe Thomas, Robert Prückl, Stefan Schaffelhofer, Stefan Treue.

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
