## [Decision Letter · Decision Letter 0]

24 Mar 2024

Using camera-guided electrode microdrive navigation for precise 3D targeting of macaque brain sites

PONE-D-24-01916

Dear Dr. Crayen,

We’re pleased to inform you that your manuscript has been judged scientifically suitable for publication and will be formally accepted for publication once it meets all outstanding technical requirements.

Kind regards,

Uma Maheswari Rajagopalan, Ph.D

Academic Editor

PLOS ONE

2. To comply with PLOS ONE submissions requirements, in your Methods section, please provide additional information regarding the experiments involving animals and ensure you have included details on methods of anesthesia and/or analgesia.

“This study was supported by a grant to ST (Deutsche Forschungsgemeinschaft (DFG, German Research Foundation) – Projektnummer 436260547).”

Please respond by return e-mail so that we can amend your financial disclosure and competing interests on your behalf.

5. We note that Figures 1 and 3 in your submission contain copyrighted images. All PLOS content is published under the Creative Commons Attribution License (CC BY 4.0), which means that the manuscript, images, and Supporting Information files will be freely available online, and any third party is permitted to access, download, copy, distribute, and use these materials in any way, even commercially, with proper attribution. For more information, see our copyright guidelines: http://journals.plos.org/plosone/s/licenses-and-copyright.

1. You may seek permission from the original copyright holder of Figures 1 and 3 to publish the content specifically under the CC BY 4.0 license.

Reviewers' comments:

Reviewer's Responses to Questions

**Comments to the Author**

1. Is the manuscript technically sound, and do the data support the conclusions?

Reviewer #1: Yes

2. Has the statistical analysis been performed appropriately and rigorously? 

Reviewer #1: Yes

3. Have the authors made all data underlying the findings in their manuscript fully available?

Reviewer #1: Yes

4. Is the manuscript presented in an intelligible fashion and written in standard English?

Reviewer #1: Yes

5. Review Comments to the Author

Reviewer #1: Crayen et al. tested the spatial accuracy and practicality of their newly developed camera-based system for real-time 3D electrode tip position estimation in the cerebral cortex. First, they tested the spatial accuracy of the system on a degenerated egg white brain mockup and an actual macaque monkey brain, both with good results. Next, the system was applied to macaque MT retinotopic mapping to prove its utility. These tests were thorough, and the advantages of this method are evident. This system will be helpful to neuroscience research in awake macaque and deep brain stimulation in patients. Therefore, the publication of this paper is acceptable.

6. PLOS authors have the option to publish the peer review history of their article (what does this mean?). If published, this will include your full peer review and any attached files.

Reviewer #1: No

---

## [Editor Report · Acceptance letter]

17 May 2024

PONE-D-24-01916 

PLOS ONE

Dear Dr. Crayen, 

I'm pleased to inform you that your manuscript has been deemed suitable for publication in PLOS ONE. Congratulations! Your manuscript is now being handed over to our production team.

Kind regards, 

on behalf of

Dr. Uma Maheswari Rajagopalan 

Academic Editor

PLOS ONE